:PLOS | ONE

# Modulations of microbehaviour by associative memory strength in *Drosophila* larvae

**Michael Thane**[☉], **Vignesh Viswanathan**[¤☉], **Tessa Christin Meyer, Emmanouil Paisios, Michael Schleyer**[ID]*

Leibniz Institute for Neurobiology (LIN), Department Genetics of Learning and Memory, Magdeburg, Germany

☉ These authors contributed equally to this work.
¤ Current address: University of Marburg, Institute for Physiology and Pathophysiology, Marburg, Germany
* michael.schleyer@lin-magdeburg.de

**Data Availability Statement:** Source data to all behavioural experiments are included as supplementary material.

## Abstract

Finding food is a vital skill and a constant task for any animal, and associative learning of food-predicting cues gives an advantage in this daily struggle. The strength of the associations between cues and food depends on a number of parameters, such as the salience of the cue, the strength of the food reward and the number of joint cue-food experiences. We investigate what impact the strength of an associative odour-sugar memory has on the microbehaviour of *Drosophila melanogaster* larvae. We find that larvae form stronger memories with increasing concentrations of sugar or odour, and that these stronger memories manifest themselves in stronger modulations of two aspects of larval microbehaviour, the rate and the direction of lateral reorientation manoeuvres (so-called head casts). These two modulations of larval behaviour are found to be correlated to each other in every experiment performed, which is in line with a model that assumes that both modulations are controlled by a common motor output. Given that the *Drosophila* larva is a genetically tractable model organism that is well suited to the study of simple circuits at the single-cell level, these analyses can guide future research into the neuronal circuits underlying the translation of associative memories of different strength into behaviour, and may help to understand how these processes are organised in more complex systems.

## Introduction

Finding food sources is a fundamental task for any animal, and being able to learn cues to guide this search is crucial. Therefore, even a simple animal such as the larva of the fruit fly *Drosophila melanogaster* with a nervous system of only about 10,000 neurons is capable of associating a cue such as an odour with food such as sugar (reviewed in [1–4]). Importantly, the strength of the associative odour-food memory is variable and can be affected by various parameters. In *Drosophila* larvae, the concentration of the food reward [5–6], the dilution of the odour [7], and the number of training cycles [8–9] have been reported to change memory strength. Analogous observations have also been made for adult *Drosophila* [10–12]. In all these cases, the animals were tested in a gradient of a previously trained odour, and their

**Funding:** This study received institutional support by the Otto von Guericke Universität Magdeburg, the Wissenschaftsgemeinschaft Gottfried Wilhelm Leibniz (WGL), the Leibniz Institute for Neurobiology (LIN), as well as grant support from the Deutsche Forschungsgemeinschaft (DFG) (GE 1091/4-1 and FOR 2705 Mushroom body) (www. dfg.de) and the European Commission grant MINIMAL (FP7 – 618045) (https://ec.europa.eu/info/research-and-innovation_en). The funders had no role in study design, data collection and analysis, decision to publish, or preparation of the manuscript.

**Competing interests:** The authors have declared that no competing interests exist.

distribution was determined after a given amount of time. After paired presentations of odour and food reward, the animals prefer the odour more than after separate, unpaired presentations of odour and food reward. The difference observed in the distribution of animals, quantified as the associative Performance Index, is used as a proxy for memory strength [8,13–16]. Only recently, efforts were made to understand the actual modulations of locomotion that underlie the distribution of the animals after training [17–18]. In this study, we refer to these behavioural modifications as "microbehaviour".

Larval chemotaxis is usually described as comprising a sequence of relatively straight runs and lateral head movements called head casts (HC) that precede changes in direction [19–24] (but see [25–27]). Thus, a larva theoretically has three ways to express a preference for an odour: it can modulate its run speed according to whether it is heading towards or away from the odour source; it can modulate its HC rate according to whether it is heading towards or away from the odour source; or it can modulate the direction of HC more towards or away from the odour source. Associative memories have been found mainly to modulate the latter two aspects of chemotaxis, HC rate and HC direction [17–18].

Here, we assay groups of larvae by associative training with *n*-amyl acetate as the odour and fructose as the reward, and inquire into the microbehavioural footprint of memories of different strength induced by varying either the reward quantity, the odour concentration, or the number of training cycles.

## Material and methods

### General

Third-instar feeding-stage larvae (*Drosophila melanogaster*) of the wildtype strain Canton-S, aged 5 days after egg laying, were used throughout. Flies were maintained on standard medium, in mass culture at 25°C, 60–70% relative humidity and a 12/12 hour light/dark cycle. We took a spoonful of food medium from a food vial, randomly selected the desired number of larvae, briefly rinsed them in tap water and started the experiment.

For behavioural experiments, larvae were trained and tested in Petri dishes of 9 cm inner diameter (Sarstedt, Nümbrecht, Germany) filled with 1% agarose (electrophoresis grade; Roth, Karlsruhe, Germany). As the sugar reward, we used D-fructose (FRU; CAS: 57-48-7; Roth, Karlsruhe, Germany), at concentrations of 0.2, 0.6 or 2 mol/L. As the odour, we used an organic ester with a pear-like smell, *n*-amyl acetate (AM; CAS: 628-63-7; Merck, Darmstadt, Germany), diluted 1:2000, 1:200 or 1:20 in paraffin oil, as it has been shown that larvae can learn about this odour very well [7,14].

### Associative odour-sugar learning

The experiments followed established protocols [28]. Odour containers were prepared by adding 10 µl of odour substance into custom-made Teflon containers (5 mm inner diameter with a lid perforated with 7 holes of 0.5-mm diameter). The Petri dishes were covered with modified lids perforated in the centre by 15 holes of 1 mm diameter to improve aeration.

For the typical odour-sugar training, approximately 15 larvae were placed in the middle of a Petri dish filled with agarose that contained 2 mol/L FRU as a reward (indicated below by '+'), and equipped with two odour containers on opposite sides, both filled with AM diluted 1:20 (this combination of stimuli, AM and FRU, is abbreviated as 'AM+'). After 2.5 min, the larvae were displaced onto a fresh Petri dish with plain, tasteless agarose, equipped with two empty containers (abbreviated as 'EM'); they also spent 2.5 min in this Petri dish. Three such 'paired' training cycles were performed, in each case using fresh Petri dishes (thus, in total six Petri dishes, three containing FRU and three tasteless, were used for the training of one group

of 15 larvae). In half of the cases training started with reward-containing Petri dishes as indicated (AM+/EM), whereas in the other half of the cases the sequence was reversed (EM/AM+). For each group of larvae trained AM+/EM (or EM/AM+), a second group was trained reciprocally by separated, 'unpaired' presentations of odour and reward. That is, larvae were placed first, for example, on a tasteless Petri dish with AM, followed by a FRU-containing Petri dish without AM (AM/EM+ training). Again, in half of the cases the sequence was reversed (EM +/AM).

In three series of experiments, one of three parameters was successively changed: either the FRU concentration was set to 0.2, 0.6 or 2 mol/L, or the odour dilution was set to 1:2000, 1:200 or 1:20, or the number of training cycles was set to 1, 2 or 3. For each condition in each experiment, we assessed about 50 to 55 groups of larvae, each containing about 15 individual animals, that were trained paired, and as many groups again that were trained unpaired.

Following training, the larvae were transferred to the middle of a test Petri dish and tested for their odour preference. One side of the test Petri dish was equipped with an AM container, and the other side with an EM container. During the 3 minutes of the test, we recorded larval behaviour using a camera (Basler acA2040-90um). These videos were used for offline analysis.

All the following calculations were carried out per test Petri dish; that is, the average behaviour of all animals on one test Petri dish over the full 3 min of testing time was calculated. Thus, the sample size for each of these behavioural scores (Eqs 1, 3, 5 and 7) equals the number of groups of larvae and is stated below each box plot (S1–S3 Figs). From pairs always consisting of one paired-trained and one unpaired-trained group, we calculated differences in behaviour after paired and unpaired training (see below). Thus, the sample size for each of these difference scores (Eqs 2, 4, 6 and 8) equals the number of pairs of groups and is stated below each box plot (Fig 1).

## Data analysis

Larval behaviour was video-tracked and analysed as described in detail in [18]. In brief, the following aspects of larval behaviour were analysed per test Petri dish:

The time all the larvae spent on either side during the 3 minutes of the test was determined. From these numbers, a preference score (Pref) was calculated as:

$$Pref = \frac{time\ spent\ (AM) - time\ spent\ (EM)}{time\ (Total)} \tag{1}$$

Thus, Pref values are constrained between 1 and -1, with positive values indicating a preference for and negative values indicating avoidance of AM.

Memory strength can be assessed by the difference in preference scores between two reciprocally trained groups (a paired-trained and an unpaired-trained group). To quantify this difference, we calculated an associative Performance Index as:

$$Performance\ Index = \frac{Pref\ (Paired) - Pref\ (Unpaired)}{2} \tag{2}$$

Thus, Performance Index values can range from 1 to -1, with positive values indicating appetitive and negative values indicating aversive memory. Next, we analysed the modulation of the head cast (HC) rate (HC per second, HC/s):

$$HC\ rate-modulation = \frac{HC/s\ (away) - HC/s\ (towards)}{HC/s\ (away) + HC/s\ (towards)} \tag{3}$$

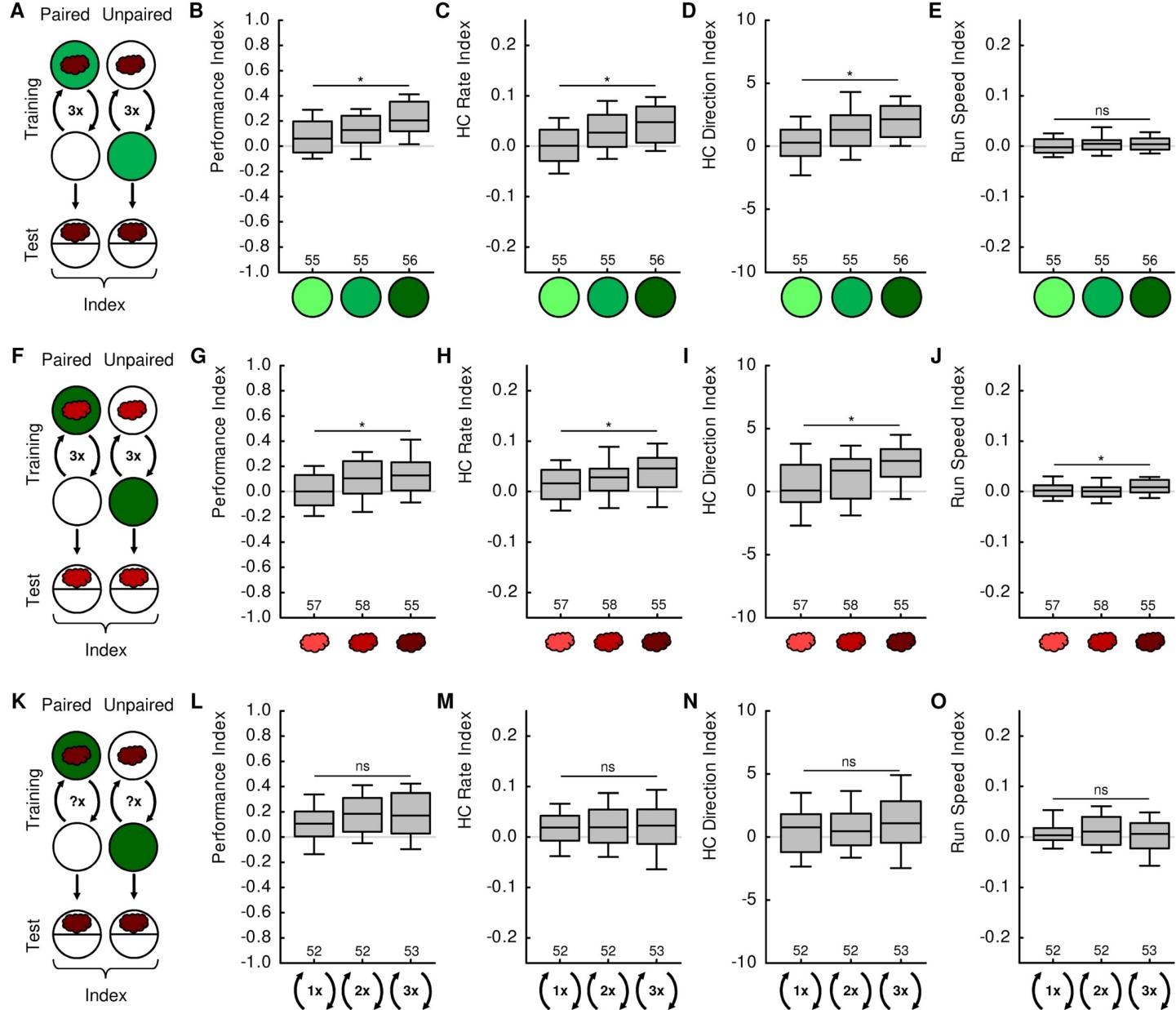

**Fig 1. Parametric modulations of learned microbehaviour.** (A) Groups of larvae were trained with either paired or unpaired presentations of *n*-amyl acetate (AM) as odour (red cloud) and fructose (FRU) as reward (green filled circles). Three training cycles were performed with an AM dilution of 1:20. The FRU concentrations used were 0.2, 0.6 and 2 mol/L, indicated by light, medium or dark green fillings in B-E, respectively. (B-D) Increasing FRU concentrations lead to increasing values of (B) the Performance Index (KW: H = 21.0, df = 2, *p* < 0.0001), (C) the HC Rate Index (KW: H = 21.6, df = 2, *p* < 0.0001), and (D) the HC Direction Index (KW: H = 23.9, df = 2, *p* < 0.0001). This indicates that higher FRU concentrations support stronger memories and lead to stronger modulations of both HC rate and HC direction. (E) FRU concentration had no significant effect on run speed modulation (KW: H = 1.5, p = 0.47). (F) As in A, except that a fixed FRU concentration of 2 mol/L was used, and AM dilutions were varied to be 1:2000, 1:200 or 1:20, indicated by light, medium or dark red clouds in G-J, respectively. (G-J) Increasing AM concentrations lead to increasing values of (G) the Performance Index (KW: H = 15.7, df = 2, *p* = 0.0004), (H) the HC Rate Index (KW: H = 8.8, df = 2, *p* = 0.012), (I) the HC Direction Index (KW: H = 15.7, df = 2, *p* = 0.0004), and (J) the Run Speed Index (KW: H = 7.9, df = 2, p = 0.020). (K) As in A, except that a fixed FRU concentration of 2 mol/L was used. 1, 2 or 3 training cycles were performed. (L-O) The number of training cycles did not significantly affect (L) the Performance Index (KW: H = 4.6, df = 2, p = 0.10), (M) the HC Rate Index (KW: H = 0.2, df = 2, p = 0.92), (N) the HC Direction Index (KW: H = 1.8, df = 2, p = 0.40), or (O) the Run Speed Index (KW: H = 0.9, p = 0.64). Thus, the memory strength was not significantly determined by training cycle number; nor were the HC rate, HC direction or run speed. Asterisks or "ns" indicate significant or non-significant Kruskal-Wallis tests, respectively. Sample sizes are indicated below each box plot. Box plots represent the median as the middle line and 25% / 75% and 10% / 90% as box boundaries and whiskers, respectively. Outliers are not displayed. The values for preference, HC rate-modulation, Reorientation per HC, and Run speed-modulation after paired and unpaired training that underlie this figure are displayed in S1–S3 Figs. For the underlying source data, see S1 Dataset.

This measure yields positive scores for attraction to the odour, which is when larvae systematically perform more HC while heading away from the odour (i.e. when odour concentration decreases) than while heading towards it (i.e. when odour concentration increases). Conversely, it yields negative scores for aversion.

To quantify the difference in HC rate-modulation between two reciprocally trained groups of animals, we calculated an associative HC Rate Index as:

$$HC \; Rate \; Index = \frac{HC \; rate{-}modulation \; (Paired) - HC \; rate{-}modulation \; (Unpaired)}{2} \quad (4)$$

Thus, positive values indicate more positive HC rate-modulations after paired than after unpaired training, and hence behaviour based on an appetitive associative memory.

The modulation of HC direction is measured by the Reorientation per HC:

$$Reorientation \; per \; HC = abs(HA \; before \; HC) - abs(HA \; after \; HC) \quad (5)$$

In this measure, the heading angle (HA) describes the orientation of the animal's head relative to the odour, with absolute heading angles (abs(HA)) of 0˚ or 180˚, for example, indicating that the odour is to the front or to the rear of the larvae, respectively. This measure thus yields positive scores for attraction to the odour, i.e. when the head cast directs the larvae towards rather than away from the odour target, whereas it yields negative scores for aversion.

To quantify the difference in HC direction between two reciprocally trained groups of animals, we calculated an associative HC Direction Index as:

$$HC \; Direction \; Index = \frac{Reorientation \; (Paired) - Reorientation \; (Unpaired)}{2} \quad (6)$$

Thus, positive values indicate a stronger bias to direct HCs towards the odour after paired than after unpaired training, and hence behaviour based on an appetitive associative memory.

Last, we also analysed the modulations of run speed:

$$Run \; speed{-}modulation = \frac{Run \; speed \; (towards) - Run \; speed \; (away)}{Run \; speed \; (towards) + Run \; speed \; (away)} \quad (7)$$

Thus, if animals modified their run speed so as to speed up whenever they headed towards the odour and slow down whenever they headed away, we would obtain a positive Run speed-modulation.

To quantify the difference in Run speed-modulation between two reciprocally trained groups of animals, we calculated an associative Run Speed Index as:

$$\begin{aligned} &Run \; Speed \; Index \\ &= \frac{Run \; speed{-}modulation \; (Paired) - Run \; speed{-}modulation \; (Unpaired)}{2} \end{aligned} \quad (8)$$

Thus, positive values would indicate stronger positive Run speed-modulations after paired than after unpaired training, and hence behaviour based on an appetitive associative memory.

## Statistical analyses

All statistical tests were performed using Statistica 11 (StatSoft, Tulsa, USA) for PC. Two-tailed non-parametric tests were used (statistical assumptions for these tests were met throughout). Values were compared across multiple groups with Kruskal-Wallis tests (KW tests), for which we state the test statistic H (sometimes reported as the $\chi^2$-value), the degrees of freedom (df) and the $p$-value. Pairwise comparisons of behaviour after paired and unpaired training were

performed with Mann-Whitney tests (MW tests). For multiple tests a Bonferroni-Holm correction was applied [29]. For correlations, the Spearman's rank correlation coefficient was used. As the software we used provides only a fixed number of decimal digits for the *p*-value, we report very small *p*-values as a maximum value (e.g. $p < 0.0001$). We present our data as box plots which represent the median as the middle line and 25% / 75% and 10% / 90% as box boundaries and whiskers, respectively. Outliers are not displayed in box plots. When scatter plots are used, all values are displayed.

## Results & discussions

How do memories of different strength affect the microbehaviour of *Drosophila melanogaster* larvae? To answer this question, we trained groups of larvae to associate an odour with a sugar reward. In three series of experiments, we successively varied one of three experimental parameters: the reward quantity, the odour concentration, or the number of training cycles. All of these parameters have previously been reported to affect the strength of the established associative memory [5–9].

### Reward quantity and odour concentration affect memory strength

First, we studied the effects of reward quantity, that is, sugar concentration (Fig 1A–1E). We trained larvae either with paired presentations of an odour with a sugar reward (Fig 1A, left), or with separate, explicitly unpaired presentations of odour and reward (Fig 1A, right). As these two groups of animals share the same experience except for the contingency between odour and sugar, any difference in behaviour in a subsequent test must be due to the difference in contingency [13]. Associative memories are therefore quantified as the associative Performance Index, i.e. the difference in odour preference after paired and after unpaired training. Stronger differences in odour preference, and thus higher Performance Index values, are usually interpreted in the literature as stronger memories [8,13–16].

For low sugar concentrations, odour preference after paired training was only slightly higher than after unpaired training (S1A Fig), resulting in a low Performance Index and indicating low levels of memory (Fig 1B). Higher sugar concentrations supported stronger memories, indicated by higher Performance Index values (Fig 1B) [5–6]. Next, we analysed three aspects of the larval microbehaviour: HC rate, HC direction and run speed. After paired training, animals made more HCs while heading away from the odour than while heading towards the odour, resulting in a positive HC rate-modulation (S1B Fig). The HC rate-modulation was higher after paired than after unpaired training (S1B Fig), as quantified by the HC Rate Index (Fig 1C). Importantly, this difference increased with increasing sugar concentration, indicated by higher HC Rate Index values (Fig 1C). Likewise, larvae biased their HC direction more towards the odour source after paired than after unpaired training (S1C Fig), as quantified by the HC Direction Index (Fig 1D). This difference too increased with increasing sugar concentration, indicated by higher HC Direction Index values (Fig 1D). Arguably, the larvae could also express an odour preference by moving faster when heading towards the odour than when heading away from the odour, which would result in a positive run speed-modulation. However, we found no such modulation either after paired or unpaired training (S1D Fig), resulting in zero Run Speed Index values for all sugar concentrations (Fig 1E). These results are in line with our previous findings that associative memories mainly modulate HC rate and HC direction, but not run speed [17–18].

Similarly, higher odour concentrations supported stronger memories (Figs 1F and 1G and S2A) [7] and increased the differences in modulations of both HC rate and HC direction after paired and unpaired training (Figs 1H and 1I and S2B and S2C). In addition, we found a very

slight increase in the differences in run speed-modulation after paired and unpaired training (Figs 1J and S2D).

Finally, we varied the number of training cycles (Figs 1K–1O and S3A–S3D). Here, we confirmed recent results showing that even a single training cycle supports a relatively low level of memory (Fig 1L, left group; S3A Fig) [9]. Additional training cycles increased memory scores only slightly and non-significantly (Fig 1L, middle and right groups). This increase was bigger and significant in previous studies [8–9]. The lack of a significant increase in memory strength was paralleled by indistinguishable modulations of HC rate, HC direction and run speed after one, two or three training cycles (Figs 1M–1O and S3B–S3D).

In summary, two of the tested parameters, reward quantity and odour concentration, supported memories of significantly varying strength. In both cases, we observed that the two major features of learned microbehaviour, HC rate and HC direction, were likewise affected: weak memories occurred together with weak modulations of HC rate and HC direction, and strong memories occurred together with strong modulations of HC rate and HC direction (Fig 1A–1J). In the third set of experiments we did not observe a significant effect of training cycle number on memory strength. Consequently, we did not find effects on the modulations of HC rate or HC direction either (Fig 1K–1O).

Regarding modulations of run speed, we only observed a significant effect of odour concentration, but not of reward quantity or trial number. One possible explanation is that in the former experiment the odour concentration during the test was also varied, whereas this parameter was constant in the latter two experiments. Different odour concentrations in the test may have different impacts on run speed. Alternatively, the observed result may hint at a generally small effect of odour-sugar memory on run speed that cannot reliably be detected across repetitions of the experiment.

## HC rate and HC direction are correlated with memory strength

Next, we asked whether the microbehaviour of a given group of animals would be correlated with memory strength (Fig 2). In line with the results described above, we observed that modulations of both HC rate (Fig 2A) and HC direction (Fig 2B) were positively correlated with memory strength, whereas run speed-modulation was not (Fig 2C). Although the observed correlations were relatively weak due to the variation across repetitions of the experiment, the same pattern was observed in every set of experiments, no matter whether we varied reward quantity (Fig 2A–2C), odour concentration (Fig 2D–2F) or trial number (Fig 2G–2I). This result is particularly noteworthy in the case of the varied number of training trials, because we did not find a significant effect of trial number on memory strength (Fig 1K–1O). Thus, although the memory strength was not significantly affected by the parameter under study, groups of animals that did happen to show a strong memory also showed strong modulations of HC rate and HC direction.

## HC rate and HC direction are correlated with each other

These findings so far support the view that odour preference is controlled by a single motor output that integrates all innate and learned behavioural tendencies [18,26–27]. In this view, larvae are thought constantly to perform left-right oscillations, the size of which is modulated by the presence of attractive or aversive stimuli. If an attractive stimulus, such as a previously rewarded odour, is to the left of the animal, oscillations to the left are increased, and oscillations to the right are decreased [26]. In our analysis, this would result in a strong bias of HCs being directed towards the odour (measured as Reorientation per HC). Likewise, if the attractive odour is to the front, the oscillations get smaller to make the larva go straight ahead, and if

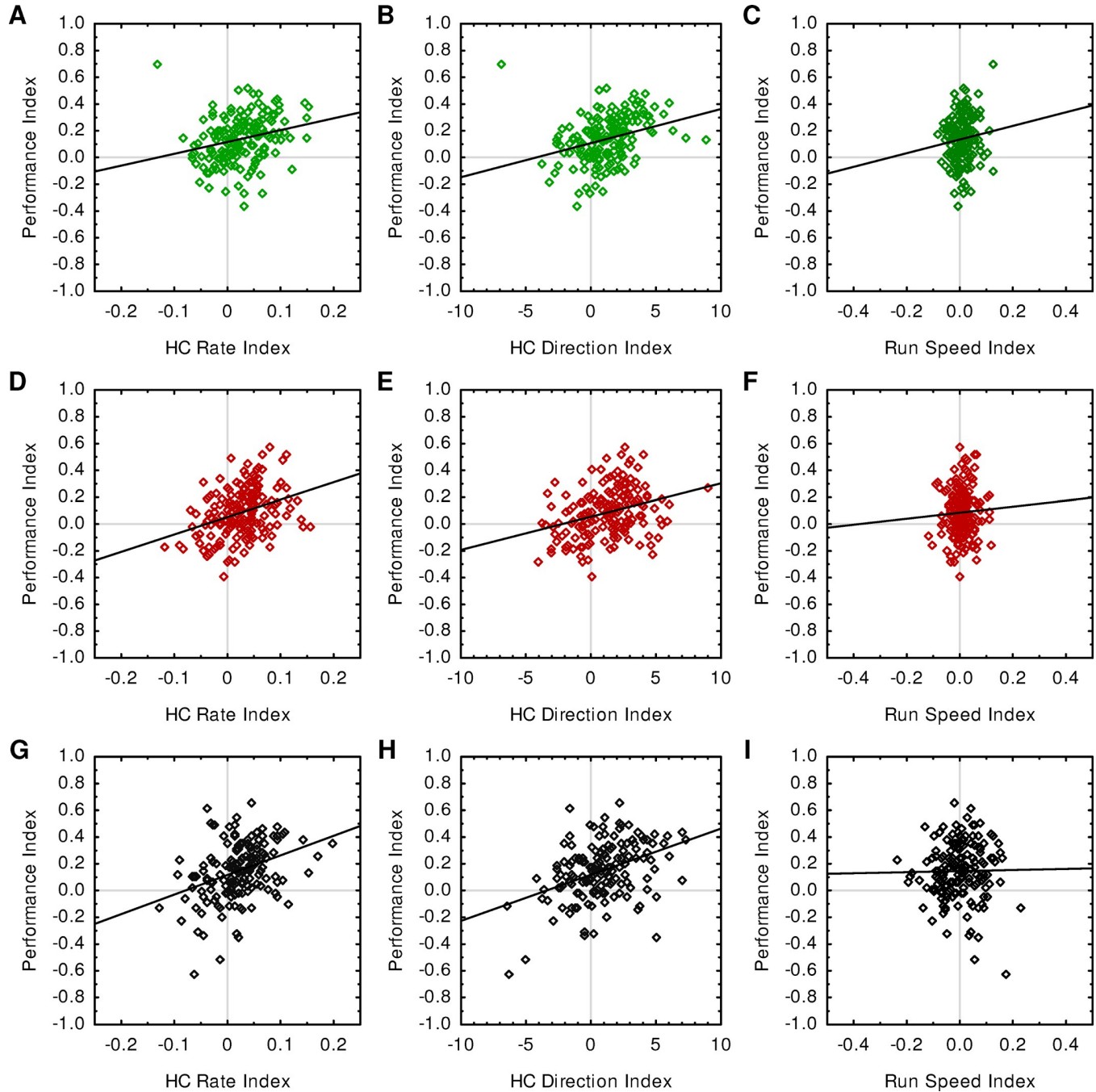

**Fig 2. Modulations of HC rate and HC direction are correlated with memory strength.** (A-C) For the experiment with varied sugar concentration, memory strength as measured by the Performance Index is positively correlated with (A) the HC Rate Index (Spearman, $r_S = 0.33$, $p < 0.00001$) and (B) the HC Direction Index (Spearman, $r_S = 0.42$, $p < 0.00001$), but not with (C) the Run Speed Index (Spearman, $r_S = 0.09$, $p = 0.228$). N = 166 each. (D-F) As in A-C, but for the experiment with varied odour concentration (Spearman, [D] $r_S = 0.33$, $p < 0.00001$; [E] $r_S = 0.32$, $p < 0.00001$; [F] $r_S = 0.05$, $p = 0.533$; N = 170 each). (G-I) As in A-C, but for the experiment with varied number of training trials (Spearman, [G] $r_S = 0.37$, $p < 0.00001$; [H] $r_S = 0.39$, $p < 0.00001$; [I] $r_S = 0.1$, $p = 0.196$; N = 156 each).

the odour is to the rear, the oscillations get bigger to make the larva turn [26]. Because our analysis uses a threshold-based detection of HCs, we detect fewer HCs when the larva is heading towards the odour, and more HCs when the larva is heading away from the odour,

resulting in a positive HC rate-modulation. Thus, according to this model HC rate and HC direction are always modulated together because both measures are merely materialisations of the same motor output.

Indeed, we found that modulations of HC rate and HC direction were positively correlated, as predicted by the model [26] (Fig 3). This was true for the HC rate-modulation and HC direction (quantified as Reorientation per HC) as calculated per group of animals (Fig 3A and 3C), as well as for the differences in HC rate-modulation (indicated by the HC Rate Index) and HC direction (indicated by the HC Direction Index) between paired-trained and unpaired-trained groups (Fig 3B and 3D). The same correlations were also observed in the case of the varied trial number, i.e. in a data set with no significant effects of the tested parameter (Fig 3E and 3F). This suggests that this correlation of modulations in HC rate and HC direction is a universal feature of larval microbehaviour, independent of the specific parameters of the experiment. However, two caveats should be kept in mind: first, the observed correlations are only weak to moderate. This may be due to technical limits of data acquisition or to high variability in the animals' behaviour, but it cannot be ruled out that HC rate and HC direction may be modulated independently of each other after all. Second, our analysis relies on group measurements. That is, we determine the average modulations of HC rate and HC direction for groups of about 15 animals. It is possible that some individuals might modulate HC rate more strongly, and others might modulate HC direction more strongly, in order to express a stronger learned odour preference. This question therefore calls for future studies analysing the behaviour of individual larvae instead of groups.

## The neuronal circuit underlying memories of different strength

The current working hypothesis for associative olfactory learning in insects locates the memory trace in the output synapses of the mushroom body Kenyon cells (reviews with focus on larvae: [1–4,30], reviews with focus on adult flies: [31–35]). The Kenyon cells receive odour information as well as reward signals which are conveyed by a subset of dopaminergic neurons (larvae: [36–39], adults: [40–44]). From the mushroom body, memory information is signalled by mushroom body output neurons towards motor control (larvae: [38–39], adults: [42,44–50]. Data from adult flies indicate that after training, the strength of the synapses from Kenyon cells to output neurons is changed, leading to a changed response in those neurons to the learned odour [47–48]. Thus, the mushroom body output neurons are thought to code the learned valence of an odour (larvae: [18,38–39], adults: [48–50]).

Our results suggest that both higher odour concentration and higher reward quantity induce stronger changes in the synaptic strength of Kenyon cell to output neuron synapses, and thus stronger deflections of the mushroom body output neurons' response to the learned odour. This results in stronger positive learned odour valence signals after paired training, and stronger negative learned odour valence signals after unpaired training. These learned odour valence signals can then be added to the odour's innate valence and accordingly shift both HC rate and HC direction more towards approach or aversion [18,26].

How the integration of innate and learned valence is organised downstream of the mushroom body output neurons is the subject of on-going research. The powerful resources at hand, such as the electron-microscopy-based connectome of the brain [38] and the genetic tools for manipulating single neurons [39], make the larva an ideal model organism for gaining a detailed circuit-level understanding of how a relatively simple nervous system translates associative memories of different strength into behaviour. The insights gained in the larva will hopefully help us also to understand how these processes are organised in more complex systems.

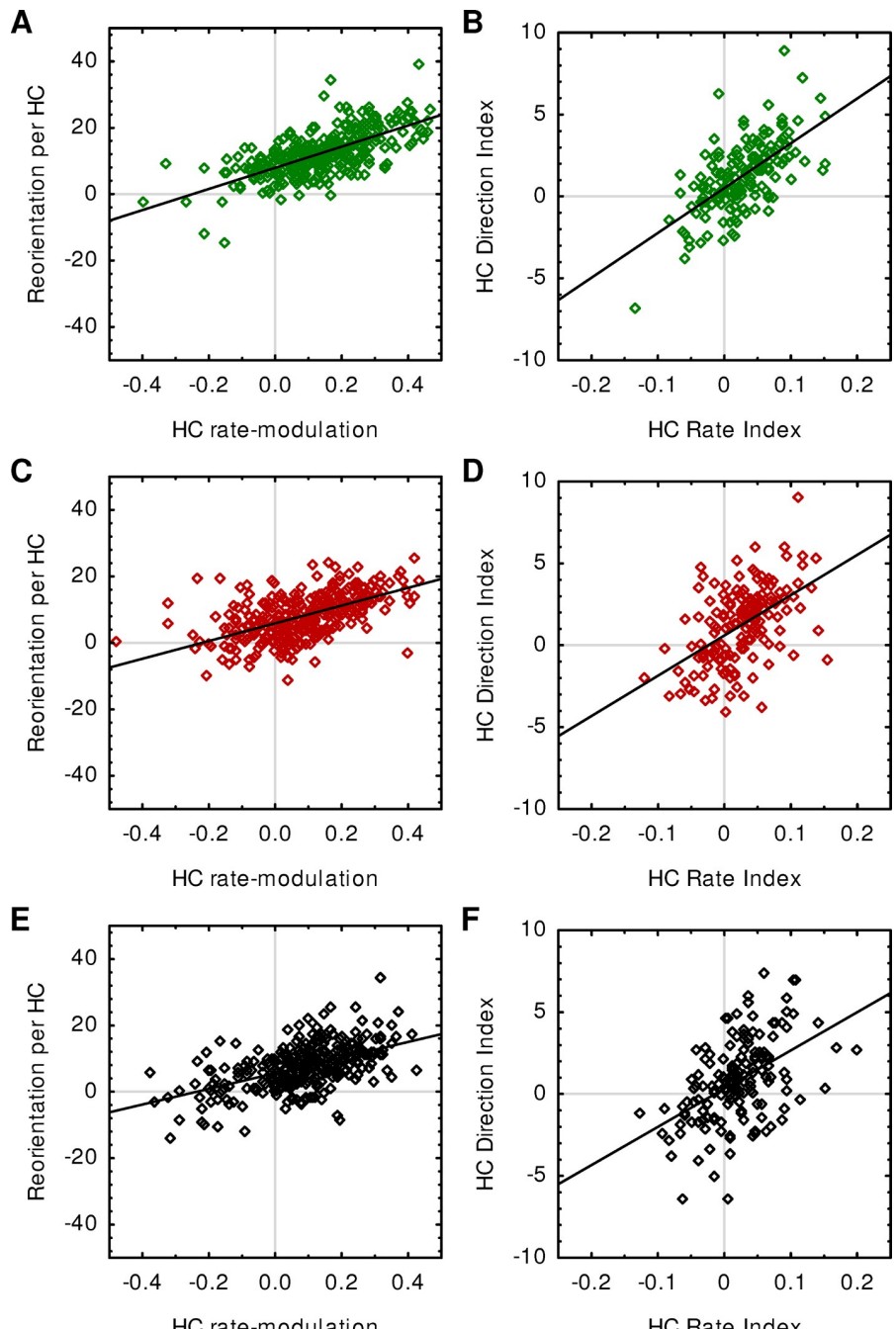

**Fig 3. Modulations of HC rate and HC direction are correlated with each other.** (A) For the experiment with varied sugar concentration, HC rate-modulation and Reorientation per HC are positively correlated (Spearman, $r_S = 0.65$, $p < 0.00001$, N = 336). This means that a group of animals that makes many HCs while heading away from the odour and few HCs while heading towards the odour (high HC rate-modulation) also shows a strong bias to direct their HCs toward the odour source (high Reorientation per HC), and *vice versa*. (B) The HC Rate Index, i.e. the difference in HC rate-modulation between a group of paired-trained and a group of unpaired-trained animals, and the HC Direction Index, i.e. the difference in Reorientation per HC between a group of paired-trained and a group of unpaired-trained animals, are positively correlated with each other (Spearman, $r_S = 0.6$, $p < 0.00001$, N = 166). (C-D) As in A-B but for the experiment with varied odour concentration (Spearman, [E] $r_S = 0.59$, $p < 0.00001$, N = 345; [F] $r_S = 0.51$, $p < 0.00001$, N = 170). (E-F) As in A-B but for the experiment with a varied number of training trials (Spearman, [G] $r_S = 0.55$, $p < 0.00001$, N = 318; [H] $r_S = 0.48$, $p < 0.00001$, N = 156).

## Supporting information

**S1 Fig. Results after paired and unpaired training with various FRU concentrations.** Displayed are the results for (A) Pref, (B) HC rate-modulation, (C) Reorientation per HC and (D) Run speed-modulation underlying the data shown in Fig 1B–1E. Results after paired training are displayed in grey-filled box plots, whereas results after unpaired training are displayed in white box plots. Asterisks indicate significant Kruskal-Wallis tests (KW) across all the paired-trained or all the unpaired-trained groups, respectively ($p < 0.05$ corrected according to Bonferroni-Holm); hash signs indicate significant Mann-Whitney U-tests (MW) between paired-trained and unpaired-trained groups ($p < 0.05$ corrected according to Bonferroni-Holm). Odour preference, HC rate-modulation and Reorientation per HC, but not Run speed-modulation, were significantly affected by the FRU concentration during paired training (KW [A] H = 15.7, df = 2, $p = 0.0004$; [B] H = 18.2, df = 2, $p = 0.0001$; [C] H = 21.1, df = 2, $p < 0.0001$; [D] H = 0.8, df = 2, $p = 0.68$). As regards unpaired training, only odour preference was significantly affected by the FRU concentration, whereas HC rate-modulation, Reorientation per HC and Run speed-modulation were not (KW [A] H = 10.3, df = 2, $p = 0.0057$; [B] H = 5.9, df = 2, $p = 0.053$; [C] H = 4.3, df = 2, $p = 0.11$; [D] H = 2.6, df = 2, $p = 0.28$). At the lowest FRU concentration, odour preference differed significantly after paired and unpaired training, but none of the aspects of chemotaxis did (MW, [A] U = 1008, $p = 0.0011$; [B] U = 1475, $p = 0.59$; [C] U = 1458, $p = 0.53$; [D] U = 1499, $p = 0.70$). Higher concentrations and therefore stronger memories correspond to significant differences after paired and unpaired training in odour preference, HC rate-modulation and Reorientation per HC, but not Run speed-modulation (MW [A] U = 721, $p = 0.00001$; U = 258, $p < 0.00001$; [B] U = 690, $p < 0.00001$; U = 521, $p < 0.00001$; [C] U = 792, $p < 0.00001$; U = 432, $p < 0.00001$; [D] U = 1281, $p = 0.10$; U = 1198, $p = 0.032$). Sample sizes are indicated below each box plot. Box plots represent the median as the middle line and 25% / 75% and 10% / 90% as box boundaries and whiskers, respectively. Outliers are not displayed.
(TIF)

**S2 Fig. Results after paired and unpaired training with various AM concentrations.** Displayed are the results for (A) Pref, (B) HC rate-modulation, (C) Reorientation per HC and (D) Run speed-modulation underlying the data shown in Fig 1G–1J. Results after paired training are displayed in grey-filled box plots, whereas results after unpaired training are displayed in white box plots.
Odour preference and Reorientation per HC, but not HC rate-modulation or Run speed-modulation, were significantly affected by the AM concentration during paired training (KW [A] H = 9.9, df = 2, $p = 0.0071$; [B] H = 3.2, df = 2, $p = 0.20$; [C] H = 9.1, df = 2, $p = 0.011$; [D] H = 2.8, df = 2, $p = 0.25$). As regards unpaired training, odour preference, HC rate-modulation and Reorientation per HC were significantly affected by the AM concentration, whereas Run speed-modulation was not (KW [A] H = 10.0, df = 2, $p = 0.0066$; [B] H = 7.9, df = 2, $p = 0.02$; [C] H = 10.8, df = 2, $p = 0.0046$; [D] H = 1.3, df = 2, $p = 0.52$). At the lowest AM concentration, only HC rate-modulation differed after paired and unpaired training (MW, [A] U = 1610, $p = 0.69$; [B] U = 1216, $p = 0.010$; [C] U = 1401, $p = 0.12$; [D] U = 1561, $p = 0.66$). At the medium AM concentration, odour preference, HC rate-modulation and Reorientation per HC, but not Run speed-modulation, were different after paired and unpaired training (MW, [A] U = 915, $p = 0.00002$; [B] U = 915, $p = 0.00002$; [C] U = 885, $p = 0.00001$; [D] U = 1672, $p = 0.95$). At the highest AM concentration, all scores were significantly different after paired and unpaired training (MW, [A] U = 611, $p < 0.00001$; [B] U = 643, $p < 0.00001$; [C] U = 489, $p < 0.00001$; [D] U = 1162, $p = 0.013$). Sample sizes are indicated below each box plot. For

further details, see S1 Fig.
(TIF)

**S3 Fig. Results after paired and unpaired training with various numbers of training cycles.** Displayed are the results for (A) Pref, (B) HC rate-modulation, (C) Reorientation per HC and (D) Run speed-modulation underlying the data shown in Fig 1L–1O. Results after paired training are displayed in grey-filled box plots, whereas results after unpaired training are displayed in white box plots.

None of the measured aspects of chemotaxis was significantly changed by varying the number of paired-training trials (KW [A] H = 1.1, df = 2, $p$ = 0.58; [B] H = 0.9, df = 2, $p$ = 0.64; [C] H = 1.0, df = 2, $p$ = 0.61; [D] H = 2.7, df = 2, $p$ = 0.26). Odour preference and Reorientation per HC were significantly affected by the number of unpaired-training trials, whereas HC rate-modulation and Run speed-modulation were not (KW [A] H = 7.5, df = 2, $p$ = 0.023; [B] H = 3.8, df = 2, $p$ = 0.15; [C] H = 8.8, df = 2, $p$ = 0.012; [D] H = 0.8, df = 2, $p$ = 0.67). Irrespective of the number of training trials, odour preference, HC rate-modulation and Reorientation per HC differed after paired and unpaired training, whereas Run speed-modulation did not (MW from left to right: [A] U = 858, $p$ = 0.00057; U = 613, $p < 0.00001$; U = 545, $p < 0.00001$; [B] U = 900, $p$ = 0.0015; U = 830, $p$ = 0.00045; U = 995, $p$ = 0.0067; [C] U = 1084, $p$ = 0.043; U = 923, $p$ = 0.0036; U = 909, $p$ = 0.0012; [D] U = 1153, $p$ = 0.11; U = 1043, $p$ = 0.032; U = 1350, $p$ = 0.62). Sample sizes are indicated below each box plot. For further details, see S1 Fig.
(TIF)

**S1 Dataset. Source data of all behavioural experiments.** This Excel file contains all the data displayed in this study. The data underlying Figs 1, 2, 3B, 3D and 3F are shown in the first tab and organised in accordance with Fig 1. The data underlying Figs S1 and 3A are shown in the second tab; the data underlying Figs S2 and 3C are shown in the third tab; and the data underlying Figs S3 and 3E are shown in the last tab.
(XLSX)

## Acknowledgments

The experimental contributions of J. Saumweber and J. Thöner, the technical assistance of M. Dombach and F. Unterstab, as well as discussions with B. Gerber, S. Gläß, C. König, M. Louis, N. Mancini, N. Toshima, B. Webb and A. Wystrach are gratefully acknowledged. We thank R. D.V. Glasgow (Zaragoza, Spain) for language editing.

## Author Contributions

**Conceptualization:** Vignesh Viswanathan, Emmanouil Paisios, Michael Schleyer.

**Data curation:** Michael Thane, Michael Schleyer.

**Formal analysis:** Michael Thane, Emmanouil Paisios, Michael Schleyer.

**Investigation:** Michael Thane, Vignesh Viswanathan, Tessa Christin Meyer, Michael Schleyer.

**Methodology:** Emmanouil Paisios.

**Project administration:** Michael Schleyer.

**Software:** Michael Thane, Emmanouil Paisios.

**Supervision:** Michael Schleyer.

**Validation:** Michael Schleyer.

**Visualization:** Michael Schleyer.

**Writing – original draft:** Michael Schleyer.

**Writing – review & editing:** Michael Thane, Vignesh Viswanathan, Tessa Christin Meyer, Michael Schleyer.

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
