## [Decision Letter · Decision Letter 0]

6 Aug 2019

PONE-D-19-18603

Modulations of microbehaviour by associative memory strength in Drosophila larvae

PLOS ONE

Dear Dr Schleyer

Thank you for submitting your manuscript to PLOS ONE. After careful consideration, we feel that it has merit but does not fully meet PLOS ONE’s publication criteria as it currently stands. Therefore, we invite you to submit a revised version of the manuscript that addresses the points raised during the review process.

First, let me apologise for the little delay in this decision, as i have been on my annual family vacation.

As you see from the attached comments both reviewers and myself agree that the manuscript has a lot of merit, but as detailed by Reviewer #1, there are some largely technical issues to be resolved that will enhance the presentation and comprehension of the manuscript.  Please thoroughly address all comments and suggestions paying particular attention to the comments related to the statistics.

We would appreciate receiving your revised manuscript by Sep 20 2019 11:59PM. To enhance the reproducibility of your results, we recommend that if applicable you deposit your laboratory protocols in protocols.io, where a protocol can be assigned its own identifier (DOI) such that it can be cited independently in the future. For instructions see: http://journals.plos.org/plosone/s/submission-guidelines#loc-laboratory-protocols

We look forward to receiving your revised manuscript.

Kind regards,

Efthimios M. C. Skoulakis, PhD

Academic Editor

PLOS ONE

Journal Requirements:

Reviewers' comments:

Reviewer's Responses to Questions

**Comments to the Author**

1. Is the manuscript technically sound, and do the data support the conclusions?

Reviewer #1: Yes

Reviewer #2: Yes

2. Has the statistical analysis been performed appropriately and rigorously? 

Reviewer #1: No

Reviewer #2: Yes

3. Have the authors made all data underlying the findings in their manuscript fully available?

Reviewer #1: Yes

Reviewer #2: Yes

4. Is the manuscript presented in an intelligible fashion and written in standard English?

Reviewer #1: No

Reviewer #2: Yes

5. Review Comments to the Author

Reviewer #1: General comments

This manuscript 'Modulations of microbehaviour by associative memory strength in Drosophila larvae' shows the associative learning of chemical cues in Drosophila larva for searching foods. This is a nice work, which clarify the usage of environmental cues by learning occurrence with model behavior; microbehaviors. The results and discussions are written well and they are supported well by results. While, the writing of materials and methods including statistics need to be improved. For now, I'm afraid that the lack of information in materials and methods can bring misunderstandings to readers and the statistics seem to contain some mistakes. I recommend authors to ask their natives colleagues to edit their English.

Here are numbers of suggestion, which might improve the manuscript.

Through manuscripts,

*Please correct the p value. It is impossible to get 'p=0'. In addition, suggest to provide x2 value instead of H value in KW test.

In introduction

Page 1 line 36-37, suggest to erase the words 'that can guide this search' and add ' to guide this search' after 'crucial'.

Page 1 line 44, suggest to remove ',' before 'and'

Page 1 line 45-47, suggest to rewrite the sentences 'From the …[8, 10-13]'. Currently it is difficult to understand this part.

Page 2 line 59, suggest to replace the words 'reward concentration' to 'reward quantity'

Then suggest to replace the words 'odour dilution' to 'odour concentration'.

Page 2 line 58-60, please provide the outline of the experiments. The short explanation about n-amyl acetate is important even it is a common chemical used in Drosophila learning: reason of selection, type of odour. This information can be provided in materials and methods section in the paragraph 'general'.

In Materials and Methods

*The repetition number is unclear through the section. How many petri dish was used (which contained 15 individuals)?

Page 2 line 72-73 and 74-75, suggest to remove 'as mentioned in the figure legends'.

Page 2-3 line 82-85, suggest to rewrite the sentences from 'for the typical… (AM+)' since the explanation about '+' next to AM/EM is not clear. Or add the sentence, for example: the presence of fructose is showed by '+'.

Page 3 line 87-92, suggest to provide more details about the unpaired test.

Additionally, suggest to add simple explanation about 'conditioning' (or around line 137-139).

Page 3 line 106, suggest to add the word '(Pref)'after 'a preference score'.

Page 3 line 117 suggest to provide the explanation about 'Stowards' and 'Saway'.

Page 4 line 128, suggest to provide the explanation of 'abs'.

Page 5 line 151, suggest to provide the number of petri dishes used for experiments to each calculations (maybe better to add the info. near each part).

Only figure legends contain the detailed data but it would be better to comment it shortly also in Materials and Methods section.

In Results & Discussion

*Suggest to add some discussions about the Drosophila associative learning on feeding of adult flies.

*Spearman r value under 0.5 does not regard as 'clear' correlation. It is taken as 'slight' correlation in general. It would be better to provide some information about the standard correlation value in insect learning in comparison with adult flies or other insect larva.

Page 5 line 165, suggest to make 'Discussion' to plural.

Page 6 line 199-201, it seems some sentences are missing. The place of quotations seems strange.

Page 7 line 301, suggest to remove 'clearly'.

In figure legends

In Fig. 1, please correct the p value. It is impossible to get 'p=0'.

Please provide x2 value not H value for KW test.

Page 6 line 242-243, suggest to replace '(G-J)' to 'G-J'.

In Fig. 2 please correct the p value.

Suggest to add N value for all parameter, not only (A) but also (B) and (C).

In Fig. 3 please correct the p value.

Reviewer #2: The ability of an animal to form memories is typically tested by assessing changes in the behavioral response after a specific training regime. In the past years the larva of the fruit fly has been consolidated as an intriguing model to study associative learning, in particular since the entire mushroom body circuit has been mapped using ssTEM reconstruction. However, to assess the ability to form memories most experimental analyses are based on simple choice assays and do not explore how the animal adapts its behavior.

In the current manuscript Thane and colleagues use high-resolution analysis of larval locomotion and navigation to understand how different strength of memories are encoded. The approach follows a previous publication the Schleyer lab (Paisios et al., 2017), but rather focusses on the stimulus strength.

Overall the manuscript is well prepared and the data well displayed. I do not see any major issues to be addressed. The findings reported in the current manuscript is certainly relevant for the field, in particular for scientists interested in how naïve or sensory experience alters navigation the manuscript will be of great interest.

My only minor comment is that currently the only data displayed in the figures (relevant for navigation) are the correlation plots and not the statistical analyses of the actual navigational decisions. I feel it would be valuable to also include these analyses in the manuscript on parameters of navigational performance.

6. PLOS authors have the option to publish the peer review history of their article (what does this mean?). If published, this will include your full peer review and any attached files.

Reviewer #1: No

Reviewer #2: Yes: Simon Sprecher

---

## [Author Response · Author response to Decision Letter 0]

26 Sep 2019

Reviewer #1: 

"Through manuscripts,

*Please correct the p value. It is impossible to get 'p=0'. In addition, suggest to provide x2 value instead of H value in KW test."

We thank the reviewer for the suggestions. As for the p = 0, the reviewer is of course right. Unfortunately, the statistics program we use does not provide more than 5 decimal digits for Kruskal-Wallis tests, and 6 for Spearman tests. In cases of very low p-values, e.g. 0.0000 is displayed. In all of these cases, we now report e.g. p < 0.0001, and explain this way of reporting in the materials & methods section (line 163-165).

As regards reporting the test statistic, we understand that sometimes H and sometimes χ2 is reported. Given that the two statistics text books we consulted (McDonald, Handbook of Biological Statistics, and Gravetter, Wallnau, Statistics for the Behavioral Sciences) both recommend reporting the test statistic as H, and given that this is also the established way of reporting Kruskal-Wallis tests in our field, we have decided to report the H-value. We explain our way of reporting the test in the materials and methods section (lines 161-162), and hope that the reviewer can accept our decision.

Throughout the manuscript, we add the degree of freedom (df), which we forgot to report in the first submission. 

"In introduction

Page 1 line 36-37, suggest to erase the words 'that can guide this search' and add ' to guide this search' after 'crucial'."

Done.

"Page 1 line 44, suggest to remove ',' before 'and'"

Done.

"Page 1 line 45-47, suggest to rewrite the sentences 'From the …[8, 10-13]'. Currently it is difficult to understand this part."

We agree that this part was not clearly written, and have rewritten it accordingly (lines 45-49).

"Page 2 line 59, suggest to replace the words 'reward concentration' to 'reward quantity'

Then suggest to replace the words 'odour dilution' to 'odour concentration'."

We follow the reviewer’s suggestion to use reward quantity and odour concentration throughout the manuscript. 

"Page 2 line 58-60, please provide the outline of the experiments. The short explanation about n-amyl acetate is important even it is a common chemical used in Drosophila learning: reason of selection, type of odour. This information can be provided in materials and methods section in the paragraph 'general'."

We now briefly describe the general principle of the experiment in the last paragraph of the introduction (lines 60-63), and have provided some information about the odour (lines 76-78). 

"In Materials and Methods

*The repetition number is unclear through the section. How many petri dish was used (which contained 15 individuals)?"

We now try to make this point clearer, and state more precisely how the sample size relates to the number of test Petri dishes and groups of larvae tested, also giving the approximate number of test Petri dishes used for analysis in lines 102-105. We think, however, that it makes most sense to state the exact sample size in the figure, not in the methods section.

"Page 2 line 72-73 and 74-75, suggest to remove 'as mentioned in the figure legends'."

Done.

"Page 2-3 line 82-85, suggest to rewrite the sentences from 'for the typical… (AM+)' since the explanation about '+' next to AM/EM is not clear. Or add the sentence, for example: the presence of fructose is showed by '+'."

We understand that this description was not clear enough. We now explain what we mean with AM, EM and + (lines 86-90).

"Page 3 line 87-92, suggest to provide more details about the unpaired test."

We follow the suggestion and describe the unpaired training in more detail (lines 96-99).

"Additionally, suggest to add simple explanation about 'conditioning' (or around line 137-139)."

We have removed “conditioning”/ ”conditioned behaviour” as we do not use these words in the rest of the text, and explain more directly what the relevant values indicate.

"Page 3 line 106, suggest to add the word '(Pref)'after 'a preference score'."

Done.

"Page 3 line 117 suggest to provide the explanation about 'Stowards' and 'Saway'."

The “s” actually belongs to HC/s, meaning head casts per second. We understand that this could very easily be misunderstood, and have clarified it.

"Page 4 line 128, suggest to provide the explanation of 'abs'."

Done.

"Page 5 line 151, suggest to provide the number of petri dishes used for experiments to each calculations (maybe better to add the info. near each part).

Only figure legends contain the detailed data but it would be better to comment it shortly also in Materials and Methods section."

We now try to make this point clearer before the beginning of the data analysis section, and state more precisely how the sample size relates to the number of test Petri dishes and groups of larvae tested, also giving the approximate number of test Petri dishes used for analysis in lines 111-118. We think, however, that it makes no sense to repeat this statement next to each calculation, as exactly the same sample size is used for all measurements within a given experimental condition.

"In Results & Discussion

*Suggest to add some discussions about the Drosophila associative learning on feeding of adult flies.

*Spearman r value under 0.5 does not regard as 'clear' correlation. It is taken as 'slight' correlation in general. It would be better to provide some information about the standard correlation value in insect learning in comparison with adult flies or other insect larva."

In order to keep the discussion concise, we have tried to focus on larval Drosophila. However, we now mention adult studies regarding the parameters that affect appetitive memory strength in the introduction (lines 43-44), and have extended our discussion of the underlying circuitries to cover adult data as well (lines 361-370).

Regarding the Spearman value, the reviewer is of course right that the correlations we observe would generally be considered relatively weak (Fig. 2) to moderate (Fig. 3). We now qualify the results accordingly, and briefly discuss the caveats to this analysis (lines 289-290, 332-336). 

"Page 5 line 165, suggest to make 'Discussion' to plural."

Done.

"Page 6 line 199-201, it seems some sentences are missing. The place of quotations seems strange."

We see that this sentence could be misleading. We now make it clearer that in this sentence we are talking about how the Performance Index is interpreted in the literature (lines 201-202).

"Page 7 line 301, suggest to remove 'clearly'."

Done.

"In figure legends

In Fig. 1, please correct the p value. It is impossible to get 'p=0'.

Please provide x2 value not H value for KW test."

Please see our first comment above.

"Page 6 line 242-243, suggest to replace '(G-J)' to 'G-J'."

Done; also for similar cases in the figure legend.

"In Fig. 2 please correct the p value."

Please see our first comment above.

"Suggest to add N value for all parameter, not only (A) but also (B) and (C)."

We now clarify that the given sample size is the same for A-C.

"In Fig. 3 please correct the p value."

Please see our first comment above.

 

Reviewer #2: 

"Overall the manuscript is well prepared and the data well displayed. I do not see any major issues to be addressed. The findings reported in the current manuscript is certainly relevant for the field, in particular for scientists interested in how naïve or sensory experience alters navigation the manuscript will be of great interest.

My only minor comment is that currently the only data displayed in the figures (relevant for navigation) are the correlation plots and not the statistical analyses of the actual navigational decisions. I feel it would be valuable to also include these analyses in the manuscript on parameters of navigational performance."

We thank the reviewer for his supportive review and his suggestion. If we understand correctly, the reviewer would like to see statistical analyses of the data presented in Fig. S1-S3. We have applied Kruskal-Wallis tests to determine whether aspects of chemotaxis change across the tested parameters, as well as Mann-Whitney tests to determine whether aspects of chemotaxis differ after paired and unpaired training.

---

## [Decision Letter · Decision Letter 1]

8 Oct 2019

Modulations of microbehaviour by associative memory strength in Drosophila larvae

PONE-D-19-18603R1

Dear Dr. Schleyer,

We are pleased to inform you that your manuscript has been judged scientifically suitable for publication and will be formally accepted for publication once it complies with all outstanding technical requirements.

With kind regards,

Efthimios M. C. Skoulakis, PhD

Academic Editor

PLOS ONE

Additional Editor Comments (optional):

Reviewers' comments:

Reviewer's Responses to Questions

**Comments to the Author**

1. If the authors have adequately addressed your comments raised in a previous round of review and you feel that this manuscript is now acceptable for publication, you may indicate that here to bypass the “Comments to the Author” section, enter your conflict of interest statement in the “Confidential to Editor” section, and submit your "Accept" recommendation.

Reviewer #1: All comments have been addressed

Reviewer #2: All comments have been addressed

2. Is the manuscript technically sound, and do the data support the conclusions?

Reviewer #1: Yes

Reviewer #2: Yes

3. Has the statistical analysis been performed appropriately and rigorously? 

Reviewer #1: Yes

Reviewer #2: Yes

4. Have the authors made all data underlying the findings in their manuscript fully available?

Reviewer #1: Yes

Reviewer #2: Yes

5. Is the manuscript presented in an intelligible fashion and written in standard English?

Reviewer #1: Yes

Reviewer #2: Yes

6. Review Comments to the Author

Reviewer #1: General comments

This manuscript 'Modulations of microbehaviour by associative memory strength in Drosophila larvae' shows the associative learning of chemical cues in Drosophila larva for searching foods. Thank authors for their corporations. I have only minor comments.

Trough manuscript, please unify the number figures after the decimal point.

Maybe 0.001?

Ex) line 257, suggest to change 'p < 0.0001' to 'p < 0.001', Line 266, 'p = 0.0004' to 'p < 0.001' and so on.

Line 67 and Line 186, suggest to change 'Drosophila melanogaster' to 'D. melanogaster'.

Line 175, suggest to remove the sentences after 'H'; '(sometimes reported as the x2-value)'.

Line 260, no df information in the result here.

Line 326, suggest to remove the underline from the word 'differences'.

Reviewer #2: The authors have addressed all points that were raised adequately. I feel the manuscript is an interesting addition in the field, it nicely highlights the quantitative features that may be extracted from "complex" behaviours in the fruit fly larva. In particularly interesting to changing external conditions or - as done here- during learning.

7. PLOS authors have the option to publish the peer review history of their article (what does this mean?). If published, this will include your full peer review and any attached files.

Reviewer #1: No

Reviewer #2: Yes: Simon Sprecher

---

## [Editor Report · Acceptance letter]

11 Oct 2019

PONE-D-19-18603R1 

Modulations of microbehaviour by associative memory strength in *Drosophila* larvae

Dear Dr. Schleyer:

I am pleased to inform you that your manuscript has been deemed suitable for publication in PLOS ONE. Congratulations! Your manuscript is now with our production department. 

With kind regards,

on behalf of

Dr. Efthimios M. C. Skoulakis 

Academic Editor

PLOS ONE